# Evaluation of Problematic Video Game Use in Adolescents with ADHD and without ADHD: New Evidence and Recommendations

**DOI:** 10.3390/bs14070524

**Published:** 2024-06-24

**Authors:** Manuel Isorna Folgar, José M. Faílde Garrido, María D. Dapía Conde, Fátima Braña Rey

**Affiliations:** Department of Psycho-Socio-Educational Analysis and Intervention, Faculty of Education Sciences and Social Work, University of Vigo, 32004 Ourense, Spain; isorna.catoira@uvigo.gal (M.I.F.); ddapia@uvigo.gal (M.D.D.C.); fatimab@uvigo.gal (F.B.R.)

**Keywords:** adolescents, ADHD, video games, problematic video game use, video game addiction disorder

## Abstract

Video game addiction among adolescents, particularly those with ADHD, is a significant concern. To gather more insights into video game usage patterns in this population, we investigated levels of potentially problematic use, passion, motivations, and emotional/behavioral symptoms in adolescents with and without ADHD. Our cross-sectional, multicenter study involved 2513 subjects (Age M = 15.07; SD = 2.82) from 24 schools in Galicia (Spain), including 158 (6.3%) diagnosed with ADHD. We employed an ad hoc questionnaire covering sociodemographic data and ADHD diagnosis, the Questionnaire of Experiences Associated with Video Games (CERV), the scale of passion for video games, reasons for playing video games Questionaire (VMQ), and emotional/behavioral problems assessment (SDQ). Results indicated heightened vulnerability in adolescents with ADHD, manifesting in longer usage durations and higher problematic use scores. Interestingly, ADHD medication did not affect internet gaming disorder development. Motivations for gaming differed between groups, with the ADHD cohort showing distinctions in cognitive development, coping, and violent reward. Additionally, the ADHD group exhibited greater emotional/behavioral symptoms, hyperactivity, and reduced prosocial behavior.

## 1. Introduction

In recent decades, the use of video games has become a widespread leisure practice among adolescents. There were approximately 3.26 billion players in 2023, with 36% aged between 18 and 34 and 24% under the age of 18 [1]. Its widespread use, both in terms of the number of users and the time consumed, has caused progressive attention and concern, reaching the consideration of behavior addiction. It was included in section III of the DSM-5 [2] as “internet gaming disorder” (IGD), although it is advised that more research is needed [3], and as “gaming disorder” (GD) in the International Classification of Diseases-11 -ICD-11- [4]. The difference between the DSM-5 and ICD-11 classifications is that the DSM-5 provides a definition focused on psychopathology and only considers online video games, while the ICD-11 focuses on the impact on health and impact on life and makes a differentiation between online and offline video games [5].

Studies report different data on the prevalence rates of game addiction. Recent meta-analyses [6,7] from research in different countries confirm these discrepancies. Stevens et al. [6] concluded that the global prevalence is 3.05%, which drops to 1.96% if more stringent sampling investigations are taken into account. The results by Kim et al. [7] support this trend, with slight variations, identifying that the overall prevalence is 3.3%, decreasing to 2.4% in studies with representative samples. The differences, found according to the authors, are explained mainly by the research methodology used, both by the measuring instruments and by the type of sampling. There was also an increase in cases of video game addiction among students during the COVID-19 pandemic [8].

In Spain, according to the Spanish Observatory of Drugs and Addictions [9], 85.1% of students aged 14–18 declared having played in the last 12 months, with 7.1% presenting a possible IGD. This percentage rose to 12.6% when the reference population is that of students who had played video games in the last 12 months. Another report prepared by UNICEF Spain found that 16.7% of Spanish adolescents have a possible IGD, and an additional 3.1% present symptoms of a possible addiction. These figures rise to 37.7% and 8.1% among those who play every day [10]. In children and adolescents, they found that 16.8% show indications that can be associated with a future addiction, of which 13.5% are identified with “problematic use” and 3.3% with addiction [11].

On the other hand, Attention Deficit Hyperactivity Disorder (ADHD) is a neurodevelopmental disorder that affects children and adolescents around the world; it is one of the most commonly diagnosed disorders, having a prevalence ranging between 5.9 and 7.1% [12]. The behavior pattern of ADHD is defined by the core areas of inattention, hyperactivity, and impulsivity and the symptoms reported in the DSM-V associated with them. From their combination, Delgado et al. [13] proposed the following classification: combined (presence of the three symptoms), predominantly inattentive, and hyperactive–impulsive. In addition, the literature confirms the comorbidity of ADHD with other learning disorders, such as dyslexia or dysgraphia [14,15], as well as other mental disorders [16,17,18]. However, there are few studies that address the prevalence of IGD and/or GD in subjects diagnosed with ADHD; in general, they agree in associating a higher prevalence of these addictions with ADHD [19,20], although the direction of this causality is not clear, with some studies suggesting a bidirectional relationship [21], indicating greater vulnerability in children with ADHD to develop addictive behaviors in the use of video games, as well as a correlation between the severity of ADHD symptoms and excessive use of video games [22]. In the research developed by Tiraboschi et al. [23], it was shown that greater involvement and more time spent playing video games can be considered a risk factor for developing ADHD symptoms. Mathews et al. [24] concluded in a reverse direction, stating that gamers with higher severity of ADHD are at higher risk of developing a video game addiction disorder. On the contrary, Suksamaia et al. [25], in their study, although they identified a higher prevalence of video game addiction in adolescents diagnosed with ADHD, they did not find a significant association between the severity of ADHD symptoms and video game addiction.

Thus, the link between ADHD and video games is not conclusive, and some authors find no association between IGD and ADHD [26]; other studies suggest that “serious games” based on gamification may have therapeutic benefits for improving the symptomatology of ADHD [27]; we also find studies that suggest a lack of clinical evidence that children with neurodevelopmental disorders can benefit from video games [28].

IGD is considered a progressive behavior with a course of chronicity that can trigger important social, physical, and mental health problems [29]. Regarding its social impact, some of the undesirable effects reported include social isolation, cessation of hobbies or activities in the company of other people, family conflicts, difficulties in interpersonal relationships [30,31], mild to severe negative effects on their psychological well-being [10,32] at the cognitive, psychophysiological, and behavior level [33], higher levels of depression, anxiety, social anxiety, and hostility [34,35], with depression being the most common symptom [29], and school failure [26,30]. 

The study of video games has been approached from different perspectives to explain their success. Fuster et al. [36] analyzed the reasons for the success of video games, highlighting four main motivations: socialization, exploration, achievement, and dissociation. In this study, they established a theoretical relationship between the motivations of socialization and exploration with adapted play and the motivations of achievement and dissociation with maladaptive play. The transition from non-problematic to problematic or addictive use occurs when a person uses them as the predominant (sometimes exclusive) means of relieving stress, anxiety, loneliness, or depression [37,38]. 

This use of video games to avoid tasks and life problems and to escape unpleasant emotional states explains the appearance and worsening of various psychological disorders [39,40,41]. 

Passion has also been used to explain the use and abuse of video games; the dual model of passion by Vallerand et al. [42,43] is one of the most used. From this perspective, passion is conceptualized as a strong desire or tendency toward the practice of an activity to which time is devoted regularly and that is considered important in the subject’s life. This dual model distinguishes two types of passion: the harmonious (PA) and the obsessive (PO). In the PA, the activity occupies an important space in the identity without being overwhelming and can be shared harmoniously with other aspects of life; on the contrary, the PO emerges from an internalization of the activity that dominates the identity of the person. The passionate activity ends up covering a disproportionate space in the identity and conflicts with other activities in the life of the individual [44].

In relation to gender, men are more likely to use video games and to be classified as gamers with addictive or potentially problematic behavior [11,45,46]. Indeed, gendered motivations for higher levels of gaming have been identified, suggesting that different interventions for men and women may be needed to design a balanced approach to gaming [47]. 

Although the main motivations for playing video games are common to all young people, children with neurodevelopmental problems have a higher average daily time spent on digital screens, while those diagnosed with ADHD spend even more time [48]. Research has highlighted several factors associated with comorbidity between ADHD and IGD [22,49,50,51,52]. The first factor indicates that the severity of ADHD symptoms predicts a greater severity of IGD; this occurs due to its associated difficulties, such as time management, prioritization of activities, or hyper-focusing of ADHD [53]. A second factor highlights that the severity of IGD can be increased by a combination of the impulsivity present in ADHD together with the achievement system and social support provided by video games [54]. ADHD people gradually begin to develop a preference for playing video games because of their educational and personal vulnerabilities. Given the symptoms and behavior expression of people diagnosed with ADHD, they tend to be rejected by others due to their impulsiveness, inattention, and disruptive behaviors, and therefore, they become more alone [55]. To mend their loneliness, they turn to the internet and video games to a greater extent, as they feel more secure and connected with others and, at the same time, quench their constant thirst for excitement and stimulation [56,57]. A third factor is related to hypofrontality in both disorders. The reduced levels of Glutamate þ glutamine could be the basis that explains the difference between the group that presents ADHD and addiction to video games and the group that presents only the diagnosis of ADHD [58]. However, we should consider that these studies are generally carried out with clinical samples, which do not allow comparisons with their groups of community counterparts, that is, the same context, geographical area, and educational center.

In this context, and to advance our understanding of video game usage and its potential risks among a vulnerable population, we conducted a descriptive and exploratory study aimed at identifying differences between adolescents with and without an ADHD diagnosis. Specifically, the study seeks to do the following:-Determine differences in usage patterns, including play time and money invested;-Examine varying motivations for playing video games;-Assess levels of problematic video game use;-Compare levels of passion for video games;-Evaluate emotional and behavioral symptomatology associated with video game use.

By addressing these objectives, the study aims to provide a comprehensive analysis of video game engagement among adolescents with and without ADHD.

## 2. Materials and Methods

### 2.1. Participants

According to data from the Department of Education of the Xunta de Galicia, a total of 89,000 Galician students were enrolled in Compulsory Secondary Education (ESO), High School, or Vocational Training (FP) during the 2020–2021 academic year. From this population, a sample was selected by means of two-stage sampling by conglomerates; for the selection of first-level units (a total of 24 centers were randomly selected, both public and private/subsidized, urban and rural in nature, and belonging to the four Galician provinces, respecting the existing quotas at the population level). For the selection of the second-level units (individuals), a sampling by quotas was used according to gender and the cycle. The final sample consisted of a total of 2513 students, 1217 of whom were female (48.4%), 1277 male (51.81%), and 19 non-binary (0.75%), aged between 12 and 19 (M = 15.07; SD = 2.82). Of these, 72.6% were in ESO, 17.9% in High School, and 9.5% in Vocational Training. The marital status of their parents was the following: married 73.5%; divorced 18.7%; widowed 2.3%; single 5.4%.

Of the total 2513 participants, 158 (6.3%) reported having a diagnosis of ADHD established by the public health system (pediatrician, neurologist, psychiatrist, or general practitioner). This represents 9% of teenage boys and 3.5% of girls, with the proportion of men being almost four times higher than that of teenage girls at 115 boys (79.3%) and 43 girls (20.7%). In addition to psychotherapeutic treatment, 58.7% (91) of ADHD patients received pharmacological treatment, and 22.15% (35) had another associated learning disorder (dyslexia and dysgraphia).

### 2.2. Instruments

Data collection was carried out using a set of instruments elaborated ad hoc based on the proposed objectives:Questionnaire of Experiences Associated with Video Games, CERV [59], an instrument validated in Spain to detect problematic and abusive use of video games. It consists of 17 items with a four-point Likert-type response format (never/almost never, sometimes, often, and almost always). Its correction facilitates obtaining a total score, as well as two scores derived from two scales: avoidance (eight items) and negative consequences (nine items). Likewise, the total scores obtained allow grouping into three groups based on the following cut-off points: without problems with the use of video games (scores between 17 and 25 points), potential problems (between 26 and 38 points), and severe problems (between 39 and 68 points). Cronbach’s alpha coefficients for the subscales are 0.87 for negative consequences and 0.86 for avoidance, with a total Cronbach’s alpha of 0.91 [59]. In our study, the Cronbach’s alpha obtained were 0.73 and 0.82 for each of the scales, respectively, and 0.89 for the global scale.Passion for video games was assessed using the Spanish version of the Passion Scale [60]. It consists of two subscales of six items, each of which assesses harmonious passion and obsessive passion, as well as five criteria items to assess the degree of passion for the activity. Each item is scored on a seven-point Likert scale, ranging from “Strongly disagree” to “Strongly agree”. The levels of internal consistency of the scale are adequate, with α = 0.81 for HP and α = 0.87 for the OP scale. Cronbach’s alpha values, in this study, were suitable for both HP (α = 0.72) and OP (α = 0.74).The Video Game Motives Questionnaire (VMQ) [61] assesses the underlying motives for playing video games and is grouped into eight categories: recreation, social interaction, coping, violent reward, competition, fantasy, cognitive development, and personalization. Moreover, with four response options from “not at all agree” to “totally agree”. Cronbach’s alpha in the sample was 0.93.To assess emotional and behavioral problems related to mental health in adolescents, the Strengths and Difficulties Questionnaire (SDQ) [62] was chosen. Taking the last 6 months as criteria, it consists of a clinical screening self-report that measures five specific indices and a general one through 25 items, five items per index, with three response options (0 = not true; 1 = somewhat true; 2 = totally true). The specific subscales measured were Emotional Symptoms, Behavior Problems, Hyperactivity, Relationship Problems and Prosocial Behavior. The Total Difficulties subscale is the sum of the first four subscales. The levels of reliability and validity for use in adolescents are adequate [63]. In this study, Cronbach’s alpha was 0.70 for the Total Difficulties scale and 0.72 for the Prosocial Behavior scale.

The previous scales were completed with the following instruments elaborated ad hoc:Sociodemographic data questionnaire: gender, age, academic year, and diagnosis of ADHD by the public health system, including all subtypes;Questionnaire on video game use patterns: days and time played; money spent per month on video games.

### 2.3. Procedure

The data were collected in the center’s own classrooms, with prior authorization from the management teams, between October 2021 and January 2022 in groups of 15–25 individuals using a questionnaire that each participant had to complete individually. The subjects were informed of the confidentiality and anonymity of the data. Both the students and their legal guardians signed an informed consent prior to their participation. The collection of information was supervised by researchers from the University of Vigo, who had experience in carrying out this type of task. Neither the families nor the adolescents received any compensation for their collaboration.

Participation was voluntary, and the time for completing the questionnaire ranged between 30 and 45 min. The collaboration and consent of both the management of the centers and the respective associations of mothers and fathers of the students were obtained. The initial number of questionnaires collected was 2617, although 104 were rejected after an exhaustive review process, either for presenting an excessive number of missing values or inconsistencies or for distorting the established quotas.

### 2.4. Ethical Aspects

This study was approved by the ethics committee of the doctoral program of the Faculty of Educational Sciences of the University of Vigo (CE-DCEC-UVIGO-2022-10-04-5109). The principles of the Helsinki Declaration and the Council of Europe Convention [64] were always complied with. The confidentiality and privacy of the data of the participants were guaranteed in accordance with the law of the General Data Protection Regulation [65].

### 2.5. Data Analysis

First, the data were filtered and coded for subsequent analysis. Next, descriptive and exploratory analyses were carried out in order to determine the characteristics of the sample by calculating the measurements of position, dispersion, and shape, thus allowing the establishment of the first results of homogeneity and the presence of anomalous elements that could produce inconsistent results. Next, a normality test was performed using the Kolmogorov–Smirnov statistic. Considering that the sample was not normally distributed, the Kruskal–Wallis test (non-parametric alternative to ANOVA) and the Mann–Whitney U test were used to perform the post hoc contrasts (two by two). Likewise, the effect size was calculated using the Hedges g statistic (values from 0.2 to 0.49 indicate the difference as small; values from 0.5 to 0.79 as moderate; values from 0.80 to 1.29 as large; and values <d 1.30 as very large).

In the case of categorical variables, contrasts were applied with the χ^2^ statistic. The results reached significance when *p* < 0.05. SPSS software (v21, IBM Corp., Armonk, NY, USA) was used to analyze the relationships between the variables studied.

## 3. Results

As can be seen in Table 1, of the total number of participants diagnosed with ADHD, 88.6% reported playing video games compared with 77.7% without this diagnosis, revealing a statistically significant association (*X*^2^ = 10.32, *p* = *0*.000). Although the set of participants indicated that they played more regularly on weekends, adolescents diagnosed with ADHD preferred to do it more frequently both on any day of the week (*X*^2^ = 9.19, *p* = 0.001) and on weekends (*X*^2^ = 325.65, *p* = 0.000).

If we take into account the variable gender and ADHD diagnosis, the trend is characterized by higher use of video games in teenage boys compared with girls, whether they report having an established diagnosis (*X*^2^ = 26.21, *p* = 0.000) or not (*X*^2^ = 8.81, *p* = 0.001).

In order to further explore the relationship between ADHD diagnosis and the use of video games, the analyses that we will describe below are limited solely to the sample of participants who reported playing video games. Of the total sample (*n* = 2513), 78.43% of the participants (*n* = 1971) reported playing video games, of which 7.10% (140) had an ADHD diagnosis established by the health system, with an association with the group of those who had invested the most money on video games in the last month (*X*^2^ = 25.97, *p* = 0.000) and those who spent more time gaming (*X*^2^ = 12.58, *p* = 0.002). Finally, gender, still confirming the general trend of gender influence in the use of video games, was reflected with greater intensity in those adolescents diagnosed with ADHD (79.3% vs. 59.8%) (Table 2).

Of the 140 players with ADHD, 30 (21.4%) presented comorbidity with another learning disorder compared with only 28 (1.5%) of the 1831 players without a diagnosis of ADHD, with significant differences between both groups (*X*^2^ = 180.31; *p* = 0.000)

The analysis of the problematic use of video games based on ADHD diagnosis (Table 3) shows statistically significant differences between both groups in all the CERV subscales. It is the participants with ADHD who showed higher scores in Psychological Dependence and Use for Avoidance (U = 96,116.00, *p* = 0.000; g+ = −0.338), Negative Consequences of Video Game Use (U = 89,269.50, *p* = 0.000; g+ = −0.482), and in General Problematic Use (U = 91,341.00, *p* = 0.000; g+ = −0.431).

Likewise, the CERV scale based on the total score allows gamers to be classified into three categories (non-problematic use, potentially problematic use, and use with severe problems). Accordingly, in this study, we observed that 51.6% of the participants without an ADHD diagnosis reported potentially problematic or severe use compared with 68.6% of those with an ADHD diagnosis, revealing a statistically significant association (*X*^2^ = 21.33, *p* = 0.000).

On the other hand, statistically significant differences were also detected in the levels of passion (Table 3), with participants with ADHD presenting significantly higher scores in both Obsessive Passion (U = 91,981.00, *p* = 0.000; g+ = −0.44), showing that the use of video games is beyond the control of the player, covering a disproportionate space in identity and conflicting with other activities in the individual’s life, and Harmonious Passion (U = 91,981.00, *p* = 0.000; g+ = −0.35), which seems to indicate that playing video games occupies an important space in the identity of these participants, although it does not always mean significant interference in other aspects of their life.

In relation to the influence of pharmacological treatment and the levels of problems caused by the use of video games (Table 4), no association was observed with the consumption of drugs.

The motivations for the use of video games appear to be different among the participants depending on whether they had been diagnosed with ADHD. Participants diagnosed with ADHD obtained significantly higher scores in three of the subscales of the Reasons for Video Game Use Questionnaire (Table 5): Cognitive Development, where ADHD perceive greater stimulation of intellectual activity during the video game (U = 90,277.00, *p* = 0.002; g+ =−0.28); Coping (U = 99,543.00, *p* = 0.002; g+ = −0.30), indicating that the ADHD group prefers to play to reduce stress and improve their mood; and Violent Reward, with participants with ADHD also reporting a greater tendency to play for enjoyment (U = 90,079.50, *p* = 0.003; g+ = −0.27). No difference was detected in the rest of the motivations, although in all of them, the scores are higher for adolescents diagnosed with ADHD.

Finally, we proceeded to analyze the emotional and behavioral symptoms based on the ADHD diagnosis, detecting statistically significant differences (Table 6). Participants with a diagnosis scored significantly higher on the following subscales: Conduct Problems (U = 114,192.50, *p* = 0.018; g+ = 0.25); Hyperactivity (U = 92,108.50, *p* = 0.000; g+ = 0.43); and Total Difficulties (U = 104,825.50, *p* = 0.000; g+ = 0.32). On the contrary, the participants without a diagnosis of ADHD scored significantly higher in Prosocial Behavior (U = 102,986.00, *p* = 0.000; g+ = 0.43).

These data seem to indicate that among adolescents who play video games, those with a recognized ADHD diagnosis present worse emotional and behavioral adjustment, and those who do not have ADHD could be defined as having more positive social behavior.

## 4. Discussion

In the present study, approximately six percent of the participants reported having a diagnosis of ADHD established by the national health system, with the proportion of teenage boys almost four times higher than that of girls, a percentage that is in line with that reported in other studies carried out in Spain [66,67]. 

If we focus solely on adolescents who play video games, we observe that almost 80% reported using them, a percentage that increases to almost 90% when they are diagnosed with ADHD, confirming the global trend observed in the scientific literature [19,21,68]. On the other hand, players with ADHD also have greater comorbidity with other learning disorders (e.g., dyslexia, dysgraphia) than their undiagnosed peers, with results like those found in general samples [17,18]. In relation to pharmacological treatment, we did not find significant differences between players diagnosed with ADHD with and without pharmacological treatment. Based on the available research, it appears that medication for ADHD does not necessarily reduce the likelihood of developing problematic or addictive video game use. Thus, we found studies that have shown that people with ADHD, whether medicated or not, continue to have high rates of video game disorder [69]. However, other studies have reported positive results from pharmacological treatment [70]. Therefore, more research is required to establish the therapeutic efficacy of pharmacological treatment targeting problematic and excessive video game playing, with the cognitive-behavioral model being, for the moment, the most promising as an application to the treatment of video game addiction [71].

In relation to the reasons for using video games, we found statistically significant differences between players with and without ADHD, with the latter obtaining higher scores in the Cognitive Development and Violent Reward dimensions of the VMQ [61]. 

Some research maintains that video games could provide benefits at the cognitive level, such as greater visual processing and better performance in visuospatial memory, among others, which would help reduce the symptoms of ADHD [24]. Likewise, it has been reported that the release of striatal dopamine, which involves brain reward circuits, during the use of video games could improve the ability to concentrate during game time, which would provide a feeling of comfort and well-being for young people with ADHD [72]. 

In relation to the preference for violent scenes or video games, several studies have shown the attraction of adolescents with ADHD to this type of content [54] since it gives them the opportunity to express hostility without restrictions. Although there is insufficient scientific evidence on the causal relationship between the use of violent video games and aggressive behavior in children and adolescents [73,74], some studies have indicated that exposure to violent video games constitutes a risk factor for aggressive behavior and repressed social behavior [75], so it must be taken into account when providing video games to children and adolescents with ADHD.

Regarding the use of video games as a coping strategy, some studies indicate that many adolescents use video games as a way of evasion or escape from their daily problems [76]. 

In this sense, video games seem to allow adolescents with ADHD to compensate for the frustrations and failures of real life with the successes and achievements they perceive while playing in a virtual world, in addition to feeling more secure and connected with others, which would explain to a large extent its appeal to this group [19,57]. In relation to the use of video games as a coping strategy, it is important to note that some authors point out that coping motives are a relevant variable when predicting addiction to video games [77]. 

On the other hand, the use of video games can also help adolescents with ADHD and problems establishing or maintaining friendships to expand their sources of social support [78], contributing positively to starting or maintaining existing friendships [79], an aspect that is frequently reported as problematic in adolescents with ADHD [80]. This would endorse the theory of the compensation hypothesis [81], which indicates that those individuals with difficulties in initiating and maintaining social interactions in real life use online video games as an attempt to meet their needs to belong to a social group.

Regarding the use of video games, it is verified that adolescents diagnosed with ADHD present significantly higher levels of prevalence of potentially problematic and severe use [22], showing significantly longer periods of play both during the weekdays and during the weekend [23], with this being more prevalent in the male sex [11,45,46]. There are several explanatory mechanisms for this attraction of adolescents with ADHD toward video games. On the one hand, their predisposition to become bored quickly, intolerance to delay, difficulties with self-control, need for intense stimulation, and difficulties in interpersonal relationships stand out [19]. These effects are counteracted by the combination provided by video games between immediacy, obtaining immediate feedback and rewards, and the social support provided. 

At the same time, risk factors for the development of addiction to video games are also typical features of ADHD—impulsivity, difficulty managing emotions, and lack of prosocial ability [82].

In relation to passion, the results show that the ADHD group presented higher scores in both harmonious passion and obsessive passion. In general, video games are used by an important part of the players to counteract the deficiencies in their daily lives [83,84]. Our results confirm that players with ADHD perceive and experience a harmonious passion during game time to a greater extent due to the high perception of the positive consequences that derive from that game time [85]. However, on the other hand, they also present a greater obsessive passion, which is manifested in greater excitement and high involvement experienced during the time invested in the video game; this interaction tends, in the ADHD group, to a greater lack of control in the activity, generating greater negative consequences [86,87]. Taken together, these results suggest that the dimensions of passion seem to play a relevant role in defining the consequences of video game use in young people with ADHD [88].

Finally, in relation to emotional and behavioral symptoms, adolescents diagnosed with ADHD exhibited higher levels of behavioral problems, hyperactivity, and total difficulties while showing lower scores in prosocial behavior. The results are in line with those found by Masi et al. [22]. In fact, according to the literature, young people with high screen use are more likely to have socio-emotional difficulties and a strong association between increased screen time and their social development [88]. Socialization difficulties are risk factors for video game addiction [89,90], and at the same time, these are increased by being reinforced through avoidance behaviors. Among young people who are not socially comfortable and experience a sense of failure in their lives, interactions through video games reduce negative feelings such as loneliness and boredom [22,91], transforming them into an emotional feeling that Alison et al. [92] have called avatar attachment, increased low self-esteem [93], and emotional regulation problems [94] or a combination of these [95]. In this way, as Griffiths [82] and Wan and Chiou [83] affirm, video games can be used by young people to counteract the deficiencies in their daily lives and, consequently, as a way of responding to the need to escape from reality and to escape from the problems they are unable to solve, with this tendency being more present in the ADHD group. Some studies, such as that by Nuyens et al. [96], showed links between impulsivity/hyperactivity and excessive participation in online games, highlighting psychological factors such as the inability to postpone reward and impulsivity. In this sense, Han et al. [70] suggest that playing video games could be a means of self-medication for children with ADHD.

## 5. Conclusions

Video games have a highly addictive power among adolescents who use or participate in them, which leads them to invest many hours on them, and this leads to clinically significant deterioration or discomfort at the family, social, and academic levels, with this being more outstanding in the ADHD group. On the other hand, playing online video games with friends can improve attention, visual-spatial skills, and working memory, but this improvement does not necessarily apply to recreational gamers with ADHD [97]. Although it should be noted that video games can represent “a very useful pedagogical vehicle” [98], for this, it is necessary that parents and teachers present a certain audiovisual literacy, specifically related to the didactic use of video games, such as following the indications of the European content classification system for video games and other types of software [PEGI] [99].

With the caveats and limitations of a cross-sectional study, our study may provide evidence of bidirectionality [100]: children with higher impulsivity and attention problems spend more time playing video games, which in turn increases attention problems and impulsivity. This would partly corroborate the attraction hypothesis (i.e., people who are impulsive or have attention problems seek out video games). On the one hand, the ADHD symptomatology itself contributes to making video games very attractive to them, while the game itself exacerbates the symptoms of adolescents with ADHD by providing an activity that continually reinforces the need for immediate gratification. 

The high frequency and long hours of playing video games often come at the expense of other forms of leisure, such as engagement in sports or cultural activities. This phenomenon raises concerns about its impact on the development of social skills, emotional regulation, and the achievement of long-term goals. According to the emotion hypothesis, video games, especially those with exciting content such as violence, may alter the perception of other activities, such as studying or cultural activities, making them less appealing by comparison. The ongoing exposure to the excitement and instant gratification of video games may influence expectations of desired stimulation, making it difficult to focus on tasks that require sustained cognitive effort. Additionally, the displacement hypothesis suggests that time spent on video games may displace time devoted to activities that promote the development of self-regulation skills [100]. These theories, while not mutually exclusive, offer different perspectives on how video game use may impact attentional issues, highlighting the need for further research in this area. For this reason, parents should limit and monitor the time they use video games and screens to develop better prosocial skills [22]. 

It is observed that the presence of problems in interpersonal relationships or deficits in social skills, as well as loneliness, are important reasons for adolescents to seclude themselves in video games. Specifically, video games are used both as a coping tool for discomfort and as a socialization tool, mainly for those adolescents diagnosed with ADHD. Problems related to the use of video games generally conceal other difficulties in the interpersonal, emotional, and/or cognitive sphere that require more extensive interventions directed at individual, family, and social aspects [101]. Subjects with ADHD may avoid personal relationships and satisfy their social needs through the virtual world. Shen and Williams [102] found that, even though socialization is a positive factor in video games, in some cases, it can develop to such an extent that players prefer digital encounters to face-to-face contact, even altering their identity [103]. These interactions can become pathological because the person settles into a false identity that provides more satisfaction than the real identity.

## 6. Limitations

One of the strengths of this study is that it has a representative and broad sample of the population under study. However, as ADHD is a disorder with a relatively low incidence in the population, between 5.9 and 7.1% [12], the sample of adolescents with ADHD is relatively small, which prevented us from performing analyses taking into account the comorbidity with other disorders that may influence the development of IGD. In addition, it is a cross-sectional study, which limits the conclusions that can be drawn from it, so it would be recommended that future studies deepen their analysis with larger samples and longitudinal or sequential designs. Furthermore, the methodological design used means that the relationships found between the variables cannot be interpreted in terms of causality. Only a longitudinal design would make it possible to establish a causal relationship and distinguish between antecedents or prognostic and consequent factors. On the other hand, a deeper reflection reveals the need to incorporate certain variables in future studies, which would allow for better analysis and understanding of the problem. These variables include socioeconomic status, age of onset of video game use, relationship with classmates, clinical factors such as the presence of other disorders or post-traumatic stress disorder, and psychological variables (personality traits, self-esteem or assertiveness), as well as of course, the educational style used by parents and the type and control of pharmacological treatment.

Finally, the fact that the data were collected in a school context and not in Primary Care or pediatric services means that the variables analyzed were self-reported, so it is impossible to know with certainty to what extent adolescents may have underestimated or overestimated the levels of video game use and the diagnosis of ADHD itself. However, other experts in addictive behaviors have already established that anonymous and confidential self-report measures have proven to be reliable [104].

## Figures and Tables

**Table 1 behavsci-14-00524-t001:** Use of video games and ADHD (Chi squared).

	NO ADHD Freq. (%)	ADHD Freq. (%)	*X* ^2^	Sig
Play Video games			10.32	0.000
Yes	1831 (77.7)	140 (88.6)		
No	524 (22.3)	18 (11.4)		
Play any day of the week			9.19	0.001
Yes	1319 (56.0)	108 (68.4)		
No	1036 (44.0)	50 (31.6)		
Often play at the weekend			8.81	0.001
Yes	1783 (75.7)	136 (86.1)		
No	572 (24.3)	22 (13.9)		

**Table 2 behavsci-14-00524-t002:** Video game players. Association with ADHD diagnosis (Chi squared).

		NO ADHD Freq (%)	ADHD Freq (%)	*X* ^2^	Sig
Gender	Men	1084 (59.8%)	111 (79.3%)	20.86	0.000
	Women	730 (40.2%)	29 (20.7%)		
Spent money on video games	Yes	468 (24.8%)	48 (32.4%)	25.97	0.000
	No	1346 (75.2%)	92 (67.6%)		
Time spent weekly	0–2 h	1179 (64.4%)	74 (52.9%)	12.58	0.002
	3–5 h	399 (21.8%)	32 (22.9%)		
	6 o more hours	253 (13.8%)	34 (24.3%)		

**Table 3 behavsci-14-00524-t003:** Problematic use and passion for video games based on being diagnosed with ADHD (U of Mann–Whiney).

	NO ADHD (*n* = 1831) M (SD)	ADHD (*n* = 140) M (SD)	U	Sig.	g+
CERV–Psychological Dependence and Avoidance	13.97 (4.09)	15.36 (4.3)	96,116.00	0.000	−0.338
CERV–Negative Consequences of Use	13.15 (3.41)	14.82 (4.06)	89,269.50	0.000	−0.482
CERV–Total Score	27.12 (7.05)	30.19 (7.93)	91,341.00	0.000	−0.431
Harmonious Passion	9.24 (2.78)	10.23 (3.12)	95,985.00	0.000	−0.35
Obsessive Passion	9.97 (2.91)	11.27 (3.38)	91,981.00	0.044	−0.44

**Table 4 behavsci-14-00524-t004:** Levels of problems in the use of video games in players diagnosed with ADHD depending on the pharmacological treatment (Chi squared).

	Medication ADHD	No-Medication ADHD	*X* ^2^	Sig
No Problems	19 (43.2%)	25 (56.8%)	5.115	0.077
Potential Problems	48 (64%)	27 (36%)		
Severe Problems	13 (61.9%)	8 (38.1%)		

**Table 5 behavsci-14-00524-t005:** Motivations for using video games. Differences according to ADHD diagnosis (Mann–Whiney U).

	NO ADHD (*n* =1831) M (SD)	ADHD (*n* = 140) M (SD)	U	Sig.	g+
Recreation	9.68 (2.71)	9.80 (2.78)	102,413.50	0.453	−0.442
Competition	7.50 (3.28)	7.81 (3.18)	100,821.00	0.323	−0.100
Cognitive Development	4.79 (3.25)	5.70 (3.20)	90,277.00	0.002	−0.28
Confrontation	5.33 (3.67)	6.45 (3.96)	99,543.00	0.002	−0.30
Interaction	5.05 (3.33)	5.45 (3.58)	100,213,5.00	0.212	−0.11
Violent Reward	3.99 (3.69)	5.00 (3.94)	90,079,5.00	0.003	−0.27
Customization	7.10 (3.72)	7.37 (3.80)	102,384.50	0.317	−0.07
Fantasy	6.10 (3.75)	6.71 (3.53)	97,140.50	0.067	−0.16

**Table 6 behavsci-14-00524-t006:** Emotional and behavioral symptoms according to the ADHD diagnosis (Mann–Whiney U) in video game players.

	NO ADHD (*n* =1831) Media (SD)	ADHD (*n* = 140) Media (SD)	U	*p*	g+
Emotional Symptoms	13.81 (4.79)	14.53 (4.68)	116,683.00	0.076	0.15
Behavior Problems	12.35 (2.96)	13.11 (3.57)	114,192.50	0.030	0.25
Relationship Problems	15.62 (2.53)	15.85 (2.94)	122,780.00	0.402	0.09
Hyperactivity	16.16 (2.97)	17.44 (3.37)	92,108.50	0.000	0.43
Prosocial Conduct	21.06 (3.29)	19.95 (3.62)	102,986.00	0.000	−0.33
Total Difficulties	57.95 (9.26)	60.93 (9.62)	104,825.50	0.000	0.32

## Data Availability

The authors of the article are willing to share data with the editor or reviewers if they wish to do so. They only have to request it by sending an email.

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
