# Peer review of "Evaluation of Problematic Video Game Use in Adolescents with ADHD and without ADHD: New Evidence and Recommendations"

_behavsci, 2024, doi:10.3390/bs14070524_

Round 1
Reviewer 1 Report
Comments and Suggestions for Authors
Overall good... a few relativley minor issues; some clarification needed and some causal assumptions made in the discussion that need to be couched as speculation.
1. Need to say more about the demographics of the sample such as age distribution. The only demographic question they report on is sex. What about ethnic mix? Any other other demographic inforation.
2. Money spent on gaming. Video games can involve a few different types of spending including purchase of the games, and micro transactions (loot boxes). Did the question specify the type of spending? Perhaps provide the question. I make this point because the microtransactions are area of growing concern.
3. The data analysis should state which analysis method use which methods. I assume the U is for the Mann-Whitney, but when were the Kolmogorov-Smirnov or the Kruskal-Wallis test used? I could not find any instance of their use; just U and chi-square.
4. They need to provide the reader with a guide to interpret "Hedges g"; in particular how to interpret the effect sizes; Is a g+ of -0.338, a large, small, or medium effect size; a reference and perhaps a statement about the impact of the effect size. Can g+ be interpreted in a manner similar to Cohen d? A citation would also help.
5. For gender, they use "men" and "women" which typically implies adults; the participants were under the age of 18 and they state that their sample is adolescent. Perhaps teenage boys & girls or avoid the age related issue with male vs. female.
6. Clearly there is a link between ADHD and playing video games and I agree that video games may be self medication (because of the highly stimulating nature). But in conclusions they assume a causal relationship and feedback loop. "The symptoms of ADHD and addiction to videogames seem to feed each other, on the one hand, the ADHD symptomatology itself contributes to making videogames very attractive to them, while the game itself exacerbates the symptoms of adolescents with ADHD by providing an activity that continually reinforces the need for immediate". There are no citations to this statement. Their evidence in their study is for the cooccurance of ADHD, but do they actually have any evidence of (1) direction of cause or (2) a feedback loop in which symptoms are exacerbated as children play video games? Is there any evidence in their study or in previous studies that video games make ADHD symptoms worse? or better for that matter? The cross sectional evidence may support it, but it is cross sectional. While a cycle is interesting speculation, it should be made clear that it is at best speculation. Instead of "seem to feed each other", they could say "perhaps feed each other"and make it clear this is speculation that needs to be further tested.
7. "On the other hand, playing online video games with friends can improve attention, visual spatial skills, and working memory, but this improvement does not necessarily apply to recreational gamers with ADHD [97]" This was a very interesting statement, but the citation was not related to the statement. Paper 97 is about about improving working memory with a computerized task, that said nothing about video games. What is the basis for thinking that recreational gamers with ADHD cannot cognitively benefit from video games. I would suggest removing this statement or backing it up with evidence. BTW I usually don't check references, but that one sounded very important so I was rather disappointed.
8. Also in the conclusions they say "The high frequency and long hours of playing video games is carried out at the expense of other leisure activities, such as sports or other cultural activities, which can interfere with the proper development of social skills, emotional self-regulation, and the achievement of long-term goals. or delay of reinforcement". Do they have any actual evidence of this either from the study or from previous studies that video gaming actually reduces invovlement in other leisure activities or do people play video games because they weren't involved in other activities in the first place. Surely that would need longitudinal evidence to show such changes. Similarly are "social skills and emotional self regulation or long term goals [...] inhibited by video game play" or are people with poorer social skills attracted to video games as a sort of self medication. The only reference given [22] is to a cross sectional paper on adhd and not a longitudinal study. So the authors need to again either provide evidence or make it clear that this is stated a speculation.
Author Response
Thank you for your positive feedback and for highlighting areas that require clarification. We appreciate your attention to detail and your constructive comments. We will address the minor issues you pointed out and provide the necessary clarifications. Additionally, we will revise the discussion section to ensure that any causal assumptions are clearly presented as speculative rather than definitive. Your input is invaluable in enhancing the quality and rigor of our paper. Thank you once again for your thoughtful review.
- In Galicia, the immigrant population is not very significant, so no such questions are asked. We do not have more socio-demographic data. We only have data on the marital status of the parents, but we have not included it in the results because it is not an objective of this study.
- The socio-demographic data include: mean age, academic year and diagnosis of ADHD by the public health system. Your parents' marital status is added: Marital status of your parents: Married 73.5%; Divorced: 18.7%; Widowed: 2.3%; Single: 5.4%
In terms of money spent on video games. We agree with the reviewer and appreciate his input but we didn't specifically ask about lotts boxes, just how much do you spend per month on video games, skins, gems, etc. It's a good question for future research.
- U Mann-Whhiney: To verify the assumption of normality and taking into account that the sample had an n>50, the Kolmogorov Smirnov statistician was chosen.
- Since the assumption of normality was not met, the use of non-parametric statistics was chosen. Taking into account that there were two contrast groups, the Mann-Whiney U test was chosen for independent samples (tables 3, 5 and 6).
- Hedges g: To calculate the effect size, the Hedges g statistic was chosen (given that the ADHD group was small). A value between 0 and 0.19 is considered to have no effect or that the difference is inconsequential. Values from 0.2 to 0.49, the difference is small; from 0.5 to 0.79, as moderate; from 0.80 to 1.29, large; and < d 1.30, very large.
- We think it is a good idea to change men and women for teenage boys and girls. We have changed it in the text.
- We fully agree with the reviewer, our claim in the conclusion is to open a door to comorbidity of both disorders, not to establish causal relationships. We have modified the introduction of the paragraph as follows: The symptoms of ADHD and addiction to videogames seem to feed each other. Although our study is cross-sectional, the findings may provide evidence of a bidirectionality [100]: children with higher impulsivity and attention problems spend more time playing video games, which in turn increases attention problems and impulsivity. This would partly corroborate the attraction hypothesis (i.e. people who are impulsive or have attention problems seek out video games). on the one hand, the ADHD symptomatology itself contributes to making videogames very attractive to them, while the game itself exacerbates the symptoms of adolescents with ADHD by providing an activity that continually reinforces the need for immediate gratification. (Reference: Gentile DA, Swing EL, Lim CG, Khoo A. Video game playing, attention problems, and impulsiveness: Evidence of bidirectional causality. Psychol Pop Media Cult. 2012;1(1):62. https://doi.org/10.1037/a0026969)
- Once again, the reviewer is absolutely right, the bibliographic citation in which this statement is contained is in the article " Salerno L, Becheri L, Pallanti S. ADHD-gaming disorder comorbidity in children and adolescents: a narrative review. Children. 2022;9(10):1528. The article reviews research on the cognitive benefits of video games, highlighting that playing online video games with friends can improve attention, visual-spatial skills, and working memory. However, it also notes that these improvements do not necessarily apply to recreational gamers with ADHD. The corresponding bibliographic citation has been changed (97).
- The high frequency and long hours on playing video games often comes at the expense of other forms of leisure, such as engagement in sports or cultural activities. This phenomenon raises concerns about its impact on the development of social skills, emotional regulation, and the achievement of long-term goals. According to the emotion hypothesis, video games, especially those with exciting content such as violence, may alter the perception of other activities, such as studying or cultural activities, making them less appealing by comparison. The ongoing exposure to the excitement and instante gratification of video games may influence expectations of desired stimulation, making it difficult to focus on tasks that require sustained cognitive effort. Additionally, the displacement hypothesis suggests that time spent on video games may displace time devoted to activities that promote the development of self-regulation skills [100]. These theories, while not mutually exclusive, offer different perspectives on how video game use may impact attentional issues, highlighting the need for further research in this área.

Reviewer 2 Report
Comments and Suggestions for Authors
The study is interesting but encounters some important limitations in this form:
the first is the total unbalance of the sample, making it impossible to draw conclusions by carrying out comparison analyses. The incidence of ADHD is low, but it is also true that this is a sample that could be representative. The second is perhaps the limited use of instruments assessing problematic video use. It is not possible to speak of addiction with the instruments used, problematic use and addiction need to be better defined, and the definition within the study is not very clear.
It should be considered that in the face of these major criticisms I discourage the publication of this article in the following form.
As minor elements I suggest:
· Better define the objectives of the study, even if it is an exploratory study.
· Unify the indication of Cronbach's alpha.
· Better explain the discussion starting from the study's hypotheses
Best Regards
Author Response
Thank you for your comments and for highlighting the limitations of our study. We appreciate your comments and understand your concern about sample imbalance.
- We recognise that the imbalance in the sample poses challenges for comparative analyses but the sample is representative of the region of Galicia (Spain) and the 6% of subjects diagnosed with ADHD is in line with other official data. Furthermore, the statistical tests used are in line with this imbalance in the N. We are aware that the low N of ADHD is a limitation and we have reflected this in the ‘limitations’ section.
- In relation to the CERV instrument, it is a questionnaire validated with a Spanish adolescent population and has been used by various research groups and its results have been published in high impact journals.
- The wording of the study objectives has been improved. They now read as follows:
In this context, and to advance our understanding of video game usage and its potential risks among a vulnerable population, we conducted a descriptive and exploratory study aimed at identifying differences between adolescents with and without an ADHD diagnosis. Specifically, the study seeks to:
- Determine differences in usage patterns, including play time and money invested.
- Examine varying motivations for playing video games.
- Assess levels of problematic video game use.
- Compare levels of passion for video games.
- Evaluate emotional and behavioral symptomatology associated with video game use.
- By addressing these objectives, the study aims to provide a comprehensive analysis of video game engagement among adolescents with and without ADHD.
- The explanation of the Hedges G-statistic has been added.
- The wording of the conclusions and discussion has been improved.

Reviewer 3 Report
Comments and Suggestions for Authors Overall, this is an excellent paper. It is well written, and includes very relevant and useful data and analysis. It contributes to our understanding a very important and developing area in youth mental health. The introduction is thorough, useful, and well referenced. The methods are well described, and they used relevant tools in this study. the results are well described, and the discussion is very good. I have some comments that are meant to be helpful. In the abstract - on line 15, the authors document that of the 2513 subjects (M = 15.07). I wasn't clear that this referred to their age until further into the paper. I suggest making this more clear/explicitly stated in the abstract. On lines 339-346, the authors discuss the fact that their data did not show benefit from ADHD medication, when there are studies out there suggesting that ADHD medication may provide benefits for gaming disorder. I think that including this point in the discussion, and adding some information/discussion to this could be very helpful to clinicians, as many of us treating youth with ADHD and gaming 'addictions' consider the fact that ADHD medicine can help to be relevant, but the authors did not find that in this study. It is possible that this relates to the self report nature of this study, or there could be relevant factors based on who takes medicine for ADHD in this geographic area (maybe access to care or funding, etc), or maybe several other factors. I think adding this to the discussion will improve the paper - to help clinicians realize that not all data show that ADHD medicine helps gaming addiction. On lines 282-285, the authors document that 51.6% of non ADHD subjects have potentially problematic or severe use, compared to 68.6% of those with an ADHD diagnosis. This is statistically significant, and clinically relevant for ADHD subjects. That said, the finding that just over half of all subjects who do not have ADHD have potentially problematic or severe use is a very important finding as well. I have been reading about and considering screen/gaming addiction as a public health issue, and your finding of 51% of the non-ADHD subjects having potentially problematic issues is very relevant. I would suggest considering adding this as a point in the discussion, and discussing the importance of this finding. Here are two relevant references about considering internet addiction from a prevention perspective (and possibly a public health perspective): Vondráčková P, Gabrhelík R. Prevention of Internet addiction: A systematic review. J Behav Addict. 2016;5(4):568-579. doi:10.1556/2006.5.2016.085 Throuvala MA, Griffiths MD, Rennoldson M, Kuss DJ. School-based Prevention for Adolescent Internet Addiction: Prevention is the Key. A Systematic Literature Review. Curr Neuropharmacol. 2019;17(6):507-525. doi:10.2174/1570159X16666180813153806 Comments on the Quality of English Language There are some spots where the English words are not ideal:- line 113: internationalization - I believe this is meant to be 'internalization'
- line 252: refer should be 'prefer'
- In Table 4, the titles should include 'ADHD' rather than 'TDAH'
Author Response
Thank you very much for your positive feedback on our paper. We are thrilled to hear that you found it to be an excellent contribution to the field of youth mental health. We appreciate your recognition of the thoroughness and relevance of our introduction, methods, results, and discussion. Your comments are highly encouraging, and we are glad that our work has met your expectations. We have proceeded to change and improve (as far as possible) your suggestions for improvement.
- The word age is added to the Abstract.
- In relation to the question of pharmacological treatment we have modified the paragraph in the following direction: Based on available research, it appears that medication for ADHD does not necessarily reduce the likelihood of developing problematic or addictive video game use. Thus we have found studies that have shown that people with ADHD, whether medicated or not, continue to have high rates of video game disorder [69]. However, other studies have reported positive results from pharmacological treatment [70]. Therefore, more research is required to establish the therapeutic efficacy of pharmacological treatment targeting problematic and excessive video game playing, with the cognitive-behavioral model being for the moment the most promising as an application to the treatment of video game addiction [71].
- In relation to the high prevalence rates found for problematic and severe use of video games in subjects with and without ADHD, we fully agree with the reviewer that they are very high and significant. We believe that this has already been stated in the discussion and conclusions. We have not discussed this issue further as we consider that it is already implicit in the conclusions. And above all so as not to go overboard in the length. But if you consider it appropriate we will do so again.
- We have modified the grammatical errors

Round 2
Reviewer 2 Report
Comments and Suggestions for Authors
Although the manuscript has improved, my perplexities regarding the scientific impact of the article remain.
The article is well done in form, but the conclusions reached cannot, in my opinion, be generalized.
Talk of new evidence is very strong compared to an extremely limited investigation.
In general, I confirm my perplexity about the publication, more for the message than for the article.
Best regards
Author Response
Dear reviewer:
First of all, thank you for your time and dedication in reading and reviewing this study for the second time.
We, the authors, believe that we have not understood concretely what aspects you consider we should change. However, in view of your initial suggestions, we have made a series of changes and clarifications to the conclusions and limitations that we believe are in line with your suggestions. We hope you like them.
We have introduced this paragraph in the conclusions to try to make it clear that this is a longitudinal design and that we should take the results with great caution :
With the caveats and limitations of a cross-sectional study, our study can provide evidence of bidirectionality…
And above all, in the limitations we have added this paragraph to show that this is a longitudinal design and that future research should control for a series of important variables that may condition the problematic use of video games and that we have not been able to cover in this study :
Furthermore, the methodological design used means that the relationships found between the variables cannot be interpreted in terms of causality. Only a longitudinal design would make it possible to establish a causal relationship and distinguish between antecedents or prognostic and consequent factors. On the other hand, a deeper reflection reveals the need to incorporate certain variables in future studies, which would allow for a better analysis and understanding of the problem, from variables such as socioeconomic level, age of onset of video game use, relationship with classmates, to variables of a clinical nature, such as the presence of other disorders or post-traumatic stress disorder, as well as psychological variables (personality traits, self-esteem or assertiveness) and of course the educational style used by parents, or the type and control of pharmacological treatment.

Round 3
Reviewer 2 Report
Comments and Suggestions for Authors
ok